# Comparing Patient Characteristics, Clinical Outcomes, and Biomarkers of Severe Asthma Patients in Taiwan

**DOI:** 10.3390/biomedicines9070764

**Published:** 2021-07-01

**Authors:** Shih-Lung Cheng, Kuo-Chin Chiu, Hsin-Kuo Ko, Diahn-Warng Perng, Hao-Chien Wang, Chong-Jen Yu, Chau-Chyun Sheu, Sheng-Hao Lin, Ching-Hsiung Lin

**Affiliations:** 1Department of Internal Medicine, Far Eastern Memorial Hospital, Taipei 22260, Taiwan; shihlungcheng@gmail.com; 2Department of Chemical Engineering and Materials Science, Yuan Ze University, Zhongli, Taoyuan 320315, Taiwan; 3Division of Chest, Department of Internal Medicine, Luodong Poh-Ai Hospital, Yilan County 265, Taiwan; chiukc1@yahoo.com.tw; 4Department of Chest Medicine, Taipei Veterans General Hospital, Taipei 22260, Taiwan; kuohsink@ms67.hinet.net (H.-K.K.); dwperng@vghtpe.gov.tw (D.-W.P.); 5Department of Internal Medicine, National Taiwan University Hospital and College of Medicine, National Taiwan University, Taipei 10617, Taiwan; haochienwang@gmail.com (H.-C.W.); jefferycjyu@ntu.edu.tw (C.-J.Y.); 6Division of Pulmonary and Critical Care Medicine, Department of Internal Medicine, Kaohsiung Medical University Hospital, Kaohsiung Medical University, Kaohsiung City 80756, Taiwan; sheucc@gmail.com; 7Division of Chest Medicine, Department of Internal Medicine, Changhua Christian Hospital, Changhua 50006, Taiwan; shenghao@gmail.com; 8Institute of Genomics and Bioformatics, National Chung Hsing University, Taichung 40227, Taiwan; 9Recreation and Holistic Wellness, MingDao Univeristy, Changhua 50006, Taiwan

**Keywords:** asthma, eosinophil

## Abstract

Purpose: To understand the association between biomarkers and exacerbations of severe asthma in adult patients in Taiwan. Materials and Methods: Demographic, clinical characteristics and biomarkers were retrospectively collected from the medical charts of severe asthma patients in six hospitals in Taiwan. Exacerbations were defined as those requiring asthma-specific emergency department visits/hospitalizations, or systemic steroids. Enrolled patients were divided into: (1) those with no exacerbations (non-exacerbators) and (2) those with one or more exacerbations (exacerbators). Receiver operating characteristic curves were used to determine the optimal cut-off value for biomarkers. Generalized linear models evaluated the association between exacerbation and biomarkers. Results: 132 patients were enrolled in the study with 80 non-exacerbators and 52 exacerbators. There was no significant difference in demographic and clinical characteristics between the two groups. Exacerbators had significantly higher eosinophils (EOS) counts (367.8 ± 357.18 vs. 210.05 ± 175.24, *p* = 0.0043) compared to non-exacerbators. The optimal cut-off values were 292 for EOS counts and 19 for the Fractional exhaled Nitric Oxide (FeNO) measure. Patients with an EOS count ≥ 300 (RR = 1.88; 95% CI, 1.26–2.81; *p* = 0.002) or FeNO measure ≥ 20 (RR = 2.10; 95% CI, 1.05–4.18; *p* = 0.0356) had a significantly higher risk of exacerbation. Moreover, patients with both an EOS count ≥ 300 and FeNO measure ≥ 20 had a significantly higher risk of exacerbation than those with lower EOS count or lower FeNO measure (RR = 2.16; 95% CI, 1.47–3.18; *p* = < 0.0001). Conclusions: Higher EOS counts and FeNO measures were associated with increased risk of exacerbation. These biomarkers may help physicians identify patients at risk of exacerbations and personalize treatment for asthma patients.

## 1. Introduction

Asthma is a heterogeneous disease with different phenotypes affecting children and adults worldwide [1,2]. The disease is chronic in its presentation, and characterized by inflammation and constriction of airways, wheezing, cough [2,3]. There are several phenotypes and endotypes with unique immunopathological mechanisms and diagnosis and symptoms often vary from patients to patients [4]. The World Health Organization estimates that asthma affects over 300 million people around the world [5].

The Global Initiative for Asthma (GINA) reports that asthma prevalence varies across the world, impacting 1–18% of populations in different countries [2]. A nationwide survey of hospitalized patients in Taiwan found that the severity of asthma increases after 18 years of age [6]. Still, researchers generally agreed that prevalence of asthma in adults is generally lower than in western countries [7]. A time-trend analysis of asthma in Taiwanese adults found that the prevalence of asthma in 2011 was approximately 12% of those entered in the insurance system [8].

Asthma in adults is associated with various morbidities, creating higher utilization of healthcare systems. In countries such as Taiwan, governments must bear the financial burden of caring for adults with asthma-related disability through their healthcare systems [8]. A study by Sun et al. (2008) examined the healthcare utilization of asthma in Taiwanese adults by studying the healthcare utilization of the national health insurance system [9]. The mean costs of hospitalizations for patients with asthma were 2.7-fold higher than those for patients without asthma. [9]. There is also an associated quality of life decrease for patients with asthma.

The goal of treatment, as outlined by the 2020 GINA guidelines, is to control the symptoms of asthma and prevent any exacerbations. Initial treatment includes the use of short-acting bronchodilators and systemic corticosteroids [2]. In later, more severe patients, these medications may be increased in intensity. However, despite developments in medication, due to the heterogenous phenotypical nature of the disease, patients still suffer from unmanaged symptoms and remain at risk of exacerbations, needing severe medical care.

Biomarkers are measurable and reproducible biological processes that often correlate with the “clinical state” of a patient [10]. Biomarkers are particularly important for respiratory diseases to understand the best treatment options and predict the progression of the disease [3]. Biomarkers have been previously used to understand personalized treatments for patients that might benefit from asthma treatment [11]. With the growing interest in precision medicine, biomarkers are already starting to be used by clinicians in the identification and treatment of patients for a variety of other diseases [1]. For asthma, however, biomarkers have not yet been commonly used to identify and track disease progression and treatment [1].

Recent studies have shown that some traits can predict future exacerbation, which includes past exacerbation history, asthma control and type 2 inflammation. Type 2 inflammation is found in approximately 50–70% of severe asthma patients [12,13] and is characterized by the accumulation of Th2 cells [14]. Immunoglobin E (IgE), fractional exhaled nitric oxide (FeNO) and blood or sputum eosinophil (EOS) are recognized biomarkers used for the identification of type 2 inflammation [2]. Knowing these biomarkers for adult asthma would help in predicting treatment effectiveness, disease prognosis, the associated healthcare resources utilizations, and costs for patients in Taiwan. This is particularly significant given that previous lack of data for adults with asthma. The goal of this study is to understand the demographic and clinical characteristics of severe asthma patients in Taiwan. Additionally, this analysis also intends to understand the association between biomarkers and exacerbations of asthma in adults.

## 2. Materials and Methods

### 2.1. Study Design and Setting

This is a multicenter retrospective cohort study to compare patient characteristics, clinical outcomes and biomarkers for severe asthma patients in Taiwan. The severe asthma patients were enrolled in outpatient visits in six hospitals during 2016–2019, including Far Eastern Memorial Hospital, National Taiwan University Hospital, Taipei Veterans General Hospital, Kaohsiung Medical University Hospital, Changhua Christian Hospital and Poh-Ai Hospital. (approval number: FEMH: IRB No. 105131-F) and conducted in accordance with the Declaration of Helsinki. All patients provided written informed consent for participation, and the study was registered at https://www.clinicaltrials. gov (NCT02871947) on 18 August 2016. Informed consent was obtained from all subjects involved in the study.

### 2.2. Study Population

#### 2.2.1. Inclusion Criteria

All patients with more than 1 year follow up in outpatient visits were enrolled. Patients requiring medium to high dose ICS and LABA or leukotriene modifier/theophylline for the previous year, or those treated with systemic corticosteroids for more than half a year to prevent from becoming uncontrolled, or those remaining uncontrolled despite this therapy were further included in the study as severe asthma patients. Patients fulfilling at least one of the following criteria are defined as uncontrolled asthma: (1) having ACQ score > 1.5 or ACT score < 19, (2) treated with two or more bursts of systemic corticosteroids, (3) having at least one hospitalization, ICU stay or mechanical ventilation in the previous year, (4) having FEV1 < 80% predicted after appropriate bronchodilator withheld and with reduced FEV1/FVC.

#### 2.2.2. Exclusion Criteria

Patients with COPD or other pulmonary disease, with lung or other end-stage cancers, with oxygen therapy for more than 15 h a day, with long-term use of non-invasive positive pressure ventilation (NIPPV), with active tuberculosis disease or other infection or without agreement were excluded.

### 2.3. Data Collection and Outcome Measurement

We have collected the following information for 1 year from medical charts: age, gender, BMI, smoking history, pre-existing comorbidities, lung function, biomarker data, treatment and frequency of serious exacerbations. Pre-existing comorbidities contained allergic rhinitis, nasal polyps, hypertension, gastroesophageal reflux disease (GERD), heart failure (HF), diabetes mellitus (DM), and atopic history. At the time of enrollment, the participants were assessed for Asthma Control Test (ACT), lung function and biomarker data including IL-5, IL-13, IL-8, IL-17, IL-33, Tryptase, Periostin, TGF Beta, TNF Alpha, vascular endothelial growth factor (VEGF), placenta growth factor (PlGF), TSLP, eosinophil (EOS) count, and fractional exhaled nitric oxide (FeNO) and Immunoglobulin E (IgE) level.

Serious exacerbations were evaluated in the current study and defined as requiring asthma-specific emergency department visits or hospitalization, or systemic steroids after entering the database.

### 2.4. Statistical Analyses

The demographic and clinical characteristics of the study subjects were described. Continuous variables were presented as mean ± standard deviation, and the difference between groups was evaluated using the independent t test. Categorical variables were presented as numbers and percentages, and the chi-squared or Fisher’s exact test was used to evaluate the difference between groups. The receiver operating characteristic (ROC) curves were plotted to determine the optimal cut-off point value for EOS counts and FeNO measure. The optimal cut-point value was then obtained using the maximum value of Youden’s index in ROC analysis. The association between exacerbation and biomarkers was analyzed by the generalized linear model and presented using the forest plot. The *p* values less than 0.05 were considered statistically significant. All analyses were performed using the SAS/Stat system for Windows, version 9.3 (SAS Institute, Cary, NC, USA).

## 3. Results

The demographic and clinical characteristics of enrolled patients are described in Table 1. A total of 132 patients were enrolled in the study. The average age of all patients was 61.47 ± 13.12 years, and 82 (62.12%) patients were female. The average BMI was 25.43 ± 4.83, suggesting enrolled patients being overweight. In terms of pre-existing comorbidities, more than 70 percent of patients had allergic rhinitis, while less than 10 percent of patients had nasal polyps and heart failure. There were 0.73 exacerbations per year. Of 132 patients, there were 52 (39.39%) patients with one or more exacerbations (exacerbators) and 80 (60.61%) patients with no exacerbations (non-exacerbators). Comparing exacerbators and non-exacerbators, there was no significant difference in demographic and clinical characteristics. The detailed medications and biomarkers of severe asthma patients were in Appendix A.

Patient biomarker data are presented in Table 2. The average Eosinophils count was 272.67 ± 272.91, FeNO Measure was 34.12 ± 23.08 and IgE level was 241.26 ± 331.20. As compared with non-exacerbators, exacerbators had significantly higher eosinophils counts (367.8 ± 357.18 vs. 210.05 ± 175.24, *p* = 0.0043) and TSLP levels (17.16 ± 13.93 vs. 11.59 ± 14.15, *p* = 0.0166), but significantly lower tryptase levels (1283.41 ± 1480.97 vs. 2270.33 ± 3228.43, *p* = 0.0191).

The optimal cut-off points for EOS and FeNO were determined by the ROC curves (Figure 1). The optimal cut-off point values were 291.76 for EOS counts and 19 for FeNO measure. We then selected 300 as the cut-off point for EOS counts and 20 for FeNO measure. There were 90 patients with EOS count < 300 and 41 patients with EOS count ≥ 300, while there were 32 patients with FeNO measure < 20 and 96 patients with FeNO measure ≥ 20. For IgE Level, there were 63 patients with IgE level < 100 and 66 patients with IgE level ≥ 100. As compared with non-exacerbators, exacerbators had a higher percentage of patients with EOS count ≥ 300 (46.15% vs. 21.52%, *p* = 0.0029) or with FeNO measure ≥ 20 (86.27% vs. 67.53%, *p* = 0.0165); however, there was no significant difference in IgE level between exacerbators and non-exacerbators (Table 3). Moreover, we found that those with EOS ≥300 and FeNO ≥20 were more commonly treated with certain medications (with ICS/LABA/LAMA, LTA, omalizumab and prednisolone).

Since both EOS count and FeNO measure showed a statically significant relationship with exacerbation, we further combined EOS count and FeNO measure as a biomarker group to evaluate its relationship with exacerbation. There were 26 patients with EOS count < 300 and FeNo measure < 20, 6 patients with EOS count ≥ 300 and FeNO measure < 20, 62 patients with EOS count < 300 and FeNo measure ≥ 20, and 33 patients with EOS count ≥ 300 and FeNO measure ≥ 20. As compared with non-exacerbators, exacerbators had a higher percentage of patients with EOS count ≥ 300 and FeNO measure ≥ 20 (43.14% vs. 14.47%, *p* = 0.0019) (Table 4).

The generalized linear models demonstrated a significant association between the biomarker group and exacerbation (Figure 2). Patients with EOS count ≥ 300 and FeNO measure ≥ 20 were associated with the worst outcome (RR = 2.16; 95% CI, 1.47–3.18; *p* = < 0.0001).

## 4. Discussion

We conducted a retrospective review study in Taiwan to analyze the clinical and demographic characteristics of asthma patients and the association between biomarkers and number of asthma exacerbations, compared with our optimal cut-off point. We found that patients with higher EOS and FeNO levels corresponded with a higher number of exacerbations. Additionally, the combination of the two biomarkers EOS count ≥ 300 and FeNO measure ≥ 20 was significantly associated with more exacerbations compared to those with a lower level of EOS and FeNO. We concentrated on these biomarkers, considering that the GINA guidelines note that FeNO is a useful biomarker for type 2 asthma, identifies airway inflammation, and is related to levels of blood eosinophils [2]. This study defines an optimal cut off point for biomarkers and highlights the need to use biomarkers in personalized medicine to improve treatment for asthma patients.

A cross-sectional matched cohort study by Price et al. (2019), used real-world evidence to understand biomarkers in predicting asthma exacerbations for patients using inhaled corticosteroids, and validates the findings of our own study [15]. The study used data from Optum Patient Care Research database (OPRC) to categorize patients based on their FeNO and EOS levels, and two different cut-off points. Price et al. (2019) found that those with high FeNO levels (≥50 ppb) and high blood eosinophil levels (≥0.300 × 10^9^ cells/L) were more likely to have an increased rate of exacerbations (RR: 3.67) compared their matched counterparts [15]. As their study pointed out, EOS and FeNO are potentially simple biomarkers to measure, and may help targeted medication for asthma patients [15]. A study by Saito et al. (2014), found that the close monitoring of FeNO levels predicted asthma exacerbations in patients with uncontrolled disease [16]. Studies like these identify the benefits of personalized treatments and monitoring. Our study identified that the optimal cut off point values were 292 for EOS and 19 for FeNo. A systematic literature review conducted by the Agency for Healthcare Research and Quality examined various cut off points for diagnosing asthma. After reviewing 43 studies, researchers found that for cut off levels of <20 ppb and 20–30 ppb, FeNO testing had a sensitivity of 0.79 and 0.64, respectively, and specificity of 0.72 and 0.81, respectively [17]. While this fits with our own cut-off points, additional research might help improve correctly using the biomarkers FeNO and identify a cut off level that can be used by all healthcare providers.

While we studied the utility of lung function in predicting exacerbation (FEV1), our study found no correlation between the two factors. In contrast, a Swedish study by Malinovschi et al. (2016), looked at the relationship between FeNO and EOS levels and lung function [18]. This study did find a relationship between biomarkers and lung function. They noted that patients with higher FeNO levels and higher EOS counts were associated having obstructed airways (FEV1). However, their study used a FeNO cut off of 20–25 ppb and 0.3 × 109/L for blood eosinophil counts (based on age) [18]. A potential reason why we did not see an association might be our more conservative cut off point.

Additionally, while we examined the roll of the blood biomarker IgE, we found no association between exacerbations and IgE. IgE, like FEV is associated with “lower lung function” and higher levels of “allergen-specific IgE are associated with risk of asthma [19].” However, IgE may be difficult to measure, as it has a short half-life compared to other “immunoglobulin isotopes” [19,20] and low sensitivity/specificity “for detecting sputum eosinophilia” [19,21]. In our study, there may have been a relationship between IgE and exacerbations, but we did not detect it.

According to the 2020 GINA report, type 2 inflammation in asthma is common in approximately 50% of patients with asthma. It is “characterized by various cytokines ((IL)-4, IL-5, and IL-3))”, and that are produced in “adaptive immune system on recognition of allergens” [2]. Type 2 inflammation often uses biomarkers to identify mild and moderate asthma. GINA reports that type 2 inflammation is characterized by “eosinophilia or increase FeNO, and may be accompanied by atopy, whereas non-Type 2 inflammation is often characterized by increased neutrophils.” [2].

While we did not group patients into type 1 and type 2 inflammation categories, we did look at treatment behavior among sub-groups (Appendix A). We found that those with EOS ≥ 300 and FeNO ≥ 20 were more commonly treated with certain medications (with ICS/LABA/LAMA, LTA, omalizumab and prednisolone), but that outcomes (number of exacerbations) of these patients were not impacted by these treatments. A potential explanation is that these patients may have “poor sensitivity” to these medications [15]. Their EOS and FeNO levels remained high despite their medication regime. This may, in turn, indicate a need for more targeted/personalized treatment that lowers their risk of future exacerbations [15]. Biologics that use the pathway for anti-interleukin (like anti IL-4R and anti IL5) can potentially reduce the exacerbations and decrease inflammatory biomarkers. Researchers have already suggested that these biomarkers can guide biologics treatment [22,23].

Higher efficiency biologics are correlated with higher levels of FeNO and EOS two other seminal studies. A trial by Castro et al. (2018) compared an anti-IL4 inhibitor (dupilumab) at various dosages with a placebo. They found that asthma exacerbations were lower among patients who received the biologic drug, approximately 48% lower for those who received either 200 mg or 300 mg. Patients who took this drug also had higher FEV1. The higher efficacy in terms of reducing exacerbations and improving lung function are correlated with higher baseline levels of FeNO (>25 ppm) and EOS counts (>300 cells per μL) [24]. Another study by Ortega et al. (2016) studied the impact of the biologic drug mepolizumab on eosinophilic asthma. The researchers found that the biologic drug reduced the average number of exacerbations per person per year from approximately 2 to 1 when compared to placebo. They highlighted that higher baseline counts of EOS (>150 cells per μL) in patients translated to more greater reductions in exacerbation with the use of drug [25]. The Castro study emphasized the efficacy of anti-IL4 agent and the Ortega study is a post-hoc analysis of data for MENSA and DREAM study for mepolizumab. These studies, similar to our study, found that patients with higher EOS and FeNO biomarkers would have more exacerbation rates, though the cut levels of these biomarkers were different. These biologics present a way forward in treatment, as patients who use them may drastically improve their quality of life.

In our study, another marker is tryptase. Patients with severe asthma with high blood eosinophil counts and low serum tryptase levels were more likely to have greater risk of exacerbation. This result was reported in our previous study [26]. The gene expression of mast cell tryptase is increased in asthmatic epithelium, especially in the type 2-high subgroup, and predicts the responsiveness to ICS [27]. The numbers of airway tissue mast cells and the concentration of bronchoalveolar lavage tryptase can determine the efficacy of ICS treatment in persistent asthma [28]. The findings of our study indicating low levels of tryptase associated with a higher risk of exacerbation implied that lower levels of serum tryptase may be linked to non-allergic type 2 inflammation or non-type 2 inflammation (ILC2-related or neutrophilic inflammation). Therefore, lower levels of serum tryptase are potentially corticosteroid-resistant and refractory to ICS/LABA treatment and associated with high risk of asthma exacerbation.

This study had several limitations. We did not control for baseline differences between the exacerbations and no exacerbations groups. Therefore, there is no matching of the cohort on underlying comorbidities, patient covaries, or baseline treatments. However, this reflects the real application of our study, considering that not all patients who have asthma will also have the same underlying conditions. Follow up for our study was only limited to one year. A longer follow up period would allow us to see long-term effects.

## 5. Conclusions

Our study demonstrated that higher EOS counts and FeNO measure were associated with an increased risk of exacerbation. Identification of these biomarkers may help physicians identify patients at risk of exacerbations and personalize treatment for asthma patients. In the future, because of the heterogenous nature of asthma in adults, the identification of biomarkers that put adults at risk of increased exacerbations may help with treatment selection by clinicians. In the future, payers and HTAs will seek the most relevant populations to make decisions about coverage with various medications. Additional studies might examine outcomes (reduction in exacerbations) based on different treatments once the patients have been identified in terms of biomarkers.

## Figures and Tables

**Figure 1 biomedicines-09-00764-f001:**
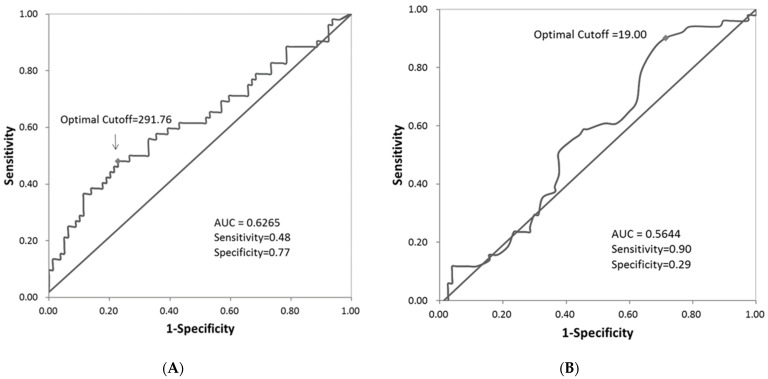
The ROC curves for the optimal cut-off point value for Eosinophils count (**A**) and FeNo measure (**B**).

**Figure 2 biomedicines-09-00764-f002:**
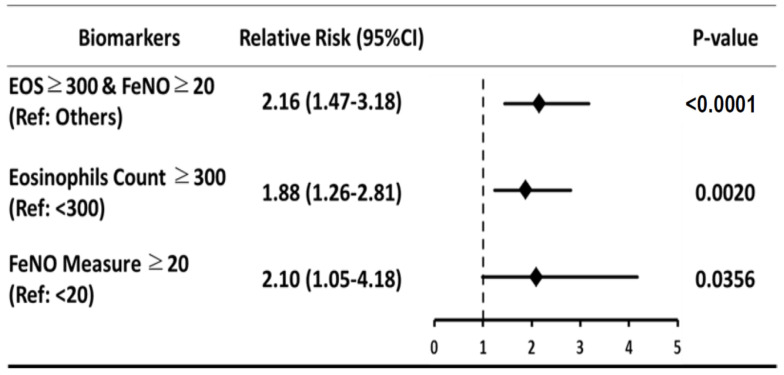
The association between Exacerbation and Biomarkers.

**Table 1 biomedicines-09-00764-t001:** Demographic and clinical characteristic of severe asthma patients.

	All Patients	No Exacerbations	Exacerbations > 1	*p*-Value
	(*n* = 132)	(*n* = 80)	(*n* = 52)	
	*n* (%)	*n* (%)	*n* (%)	
Age, mean ± SD	61.47 ± 13.12	62.19 ± 13.39	60.37 ± 12.74	0.4377
Gender				0.7980
Male	50 (37.88)	31 (38.75)	19 (36.54)	
Female	82 (62.12)	49 (61.25)	33 (63.46)	
BMI, mean ± SD	25.43 ± 4.83	25 ± 4.74	26.07 ± 4.95	0.2150
Smoking History	34 (25.76)	19 (23.75)	15 (28.85)	0.5130
Allergic Rhinitis (AR)	95 (71.97)	60 (75)	35 (67.31)	0.3363
Nasal Polyps	4 (3.03)	2 (2.5)	2 (3.85)	0.6464
Hypertension	60 (45.45)	35 (43.75)	25 (48.08)	0.6257
GERD	23 (17.42)	13 (16.25)	10 (19.23)	0.6591
Heart Failure (HF)	12 (9.09)	7 (8.75)	5 (9.62)	1.0000
DM	27 (20.45)	17 (21.25)	10 (19.23)	0.7787
Atopic	72 (54.55)	49 (61.25)	23 (44.23)	0.0550
FEV1, mean ± SD	1.54 ± 0.7	1.54 ± 0.7	1.55 ± 0.71	0.9408
FEV1%, mean ± SD	66.93 ± 23.51	68.2 ± 23.98	64.97 ± 22.85	0.4425
FEV1/FVC, mean ± SD	66.86 ± 12.14	66.82 ± 12.8	66.93 ± 11.18	0.9599
Exacerbation(s), mean ± SD	0.73 ± 1.35			

**Table 2 biomedicines-09-00764-t002:** Biomarkers of severe asthma patients.

	All Patients	No Exacerbations	Exacerbations > 1	*p*-Value
	(*n* = 132)	(*n* = 80)	(*n* = 52)	
	Mean ± SD	Mean ± SD	Mean ± SD	
IL-5	2.42 ± 1.58	2.35 ± 1.37	2.53 ± 1.87	0.5603
IL-13	59.37 ± 28.32	61.33 ± 25.7	56.39 ± 31.92	0.3528
Tryptase	1881.54 ± 2715.44	2270.33 ± 3228.42	1283.41 ± 1480.97	0.0191
Periostin	16.26 ± 10.43	15.56 ± 10.13	17.36 ± 10.9	0.3396
IL-8	15.94 ± 52.69	18.81 ± 66.55	11.41 ± 12.99	0.3403
IL-17	12.59 ± 4.4	12.68 ± 4.8	12.44 ± 3.76	0.7565
TGF Beta	30.81 ± 23.56	28.25 ± 15.3	34.86 ± 32.4	0.1816
TNF Alpha	6.17 ± 18.95	4.5 ± 11.23	8.87 ± 27.08	0.2865
VEGF	338.67 ± 296.96	346.33 ± 333.07	327.04 ± 234.27	0.6980
PIGF	7.31 ± 6.7	7.85 ± 7.23	6.47 ± 5.74	0.2563
TSLP	13.98 ± 14.33	11.59 ± 14.15	17.76 ± 13.93	0.0166
IL-33	3.87 ± 4.24	4.01 ± 4.96	3.67 ± 2.85	0.6191
EOS	272.67 ± 272.91	210.05 ± 175.24	367.8 ± 357.18	0.0043
FeNO	34.12 ± 23.08	32.77 ± 22.99	36.16 ± 23.29	0.4190
IgE Levels	241.26 ± 331.20	239.13 ± 321.43	244.52 ± 348.84	0.9283

**Table 3 biomedicines-09-00764-t003:** The relationships between exacerbation and biomarkers.

	All Patients	No Exacerbations	Exacerbations > 1	*p*-Value
	(*n* = 132)	(*n* = 80)	(*n* = 52)	
	*n* (%)	*n* (%)	*n* (%)	
Eosinophils Count				0.0029
<300	90 (68.70)	62 (78.48)	28 (53.85)	
≥300	41 (31.30)	17 (21.52)	24 (46.15)	
missing	1	1		
FeNO Measure				0.0165
<20	32 (25.00)	25 (32.47)	7 (13.73)	
≥20	96 (75.00)	52 (67.53)	44 (86.27)	
missing	4	3	1	
IgE Levels				0.9733
<100	63 (48.84)	38 (48.72)	25 (49.02)	
≥100	66 (51.16)	40 (51.28)	26 (50.98)	
missing	3	2	1	

**Table 4 biomedicines-09-00764-t004:** The relationships between exacerbation and the biomarker group.

	All Patients	No Exacerbations	Exacerbations > 1	*p*-Value
	(*n* = 132)	(*n* = 80)	(*n* = 52)	
	*n* (%)	*n* (%)	*n* (%)	
**Biomarker group #1**				0.0019
EOS <300 & FeNO < 20	26 (20.47)	20 (26.32)	6 (11.76)	
EOS ≥300 & FeNO < 20	6 (4.72)	5 (6.58)	1 (1.96)	
EOS <300 & FeNO ≥ 20	62 (48.82)	40 (52.63)	22 (43.14)	
EOS ≥300 & FeNO ≥ 20	33 (25.98)	11 (14.47)	22 (43.14)	
Missing				
**Biomarker group #2**				<0.0001
EOS ≥300 & FeNO ≥20	33 (20.47)	11 (14.47)	22 (43.14)	
Others	94 (79.53)	65 (85.53)	29 (56.86)	
Missing	5	4	1	

## Data Availability

The data will not be shared with a reason.

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
