# Peer review of "Comparing Patient Characteristics, Clinical Outcomes, and Biomarkers of Severe Asthma Patients in Taiwan"

_biomedicines, 2021, doi:10.3390/biomedicines9070764_

Round 1
Reviewer 1 Report
For a comprehensive demographic description of patients, I would also recommend information on the presence of infection, bacterial and/or viral, as a concomitant occurrence in asthma exacerbations regardless of atopic constitution.
Is there a statistical difference in both groups in terms of drug dose levels among those who did not and who had exacerbations during the quiet phase without exacerbations in both groups.
Author Response
Dear Reviewer 1:
Thanks for the review’s excellent recommendations and suggestions.
I will reply these suggestions point by point.
Best Regards.
Shih-Lung Cheng

Reviewer 2 Report
Very interesting article on biomarkers and asthma exacerbations.
Abstract:
lines 33-37 - At first these 2 sentences look to be the same. After reading the article I saw the difference. Maybe add the word both before EOS in the second sentence to draw attention to the and?
Intro:
Line 63 - this sentence seems to cut off early as if still in thought. Was the intention to add data to this sentence? Please add a reference.
Line 88 - consider adding the word biomarkers to the sentence to clarify that is where data is lacking in adults.
Line 88 - the goal of the study seems to center around biomarkers - but here this seems like a secondary goal. If this was the primary goal, please rework these 2 sentences.
Line 112: an ACT score > 19 indicates a patient is well controlled, but your study used a score > 20. Can you please explain this in the methods or limitations?
Lines 110-112: In the inclusion criteria, it mentions severe asthma patients and then uncontrolled asthma patients. How was this data used in the study? Did patients have to meet all this criteria to be included? So patients were severe and uncontrolled?
Line 129: Can you clarify if the Eos were in saliva or blood or both?
Results- can you please include how often these biomarkers were collected in patients? Once per year?
Discussion:
Line 244 - The GINA guidelines mention atopy in type 2 inflammation - but your study has less atopy in the exacerbation group. can you please discuss that?
line 246 - This data on treatment is not in the results section and should be added to results or removed from the discussion.
Line 258 - For the studies in this paragraph - do they include numbers for the EOS and FeNO levels in the biologic studies? If yes, please include so they can be compared to what you found in your patients.
Line 277 - Your sample size did detect significant differences in the 2 groups. Were you expecting to see other biomarkers impacted and that is why that sentence is included? Otherwise, can delete.
For discussion, there is no mention of the difference in Tryptase. please add to the discussion and what you think this means.
Line 168 and 170: change FeNo to FeNO
Author Response
Dear Reviewer 2:
Thanks for the review’s excellent recommendations and suggestions.
I will reply these suggestions point by point.
Best Regards.
Shih-Lung Cheng
